# EASI: Evolutionary Adversarial Simulator Identification for Sim-to-Real Transfer

**Haoyu Dong, Huiqiao Fu, Wentao Xu, Zhehao Zhou, Chunlin Chen**[*]
Department of Control Science and Intelligent Engineering,
School of Management and Engineering, Nanjing University, China
{haoyudong,hqfu, wtxu, zhzhou}@smail.nju.edu.cn,
clchen@nju.edu.cn

## Abstract

Reinforcement Learning (RL) controllers have demonstrated remarkable performance in complex robot control tasks. However, the presence of reality gap often leads to poor performance when deploying policies trained in simulation directly onto real robots. Previous sim-to-real algorithms like Domain Randomization (DR) requires domain-specific expertise and suffers from issues such as reduced control performance and high training costs. In this work, we introduce Evolutionary Adversarial Simulator Identification (EASI), a novel approach that combines Generative Adversarial Network (GAN) and Evolutionary Strategy (ES) to address sim-to-real challenges. Specifically, we consider the problem of sim-to-real as a search problem, where ES acts as a generator in adversarial competition with a neural network discriminator, aiming to find physical parameter distributions that make the state transitions between simulation and reality as similar as possible. The discriminator serves as the fitness function, guiding the evolution of the physical parameter distributions. EASI features simplicity, low cost, and high fidelity, enabling the construction of a more realistic simulator with minimal requirements for real-world data, thus aiding in transferring simulated-trained policies to the real world. We demonstrate the performance of EASI in both sim-to-sim and sim-to-real tasks, showing superior performance compared to existing sim-to-real algorithms.

## 1 Introduction

With the development of Reinforcement Learning (RL) , an increasing number of RL controllers have demonstrated remarkable performance in complex, high-dimensional robot control tasks[1, 2]. Such RL algorithms require minimal domain-specific knowledge, offering superior control performance and more natural robot locomotion compared to traditional control methods. However, training these RL controllers requires plenty of trial and error processes, it's unsafe and costly to complete training tasks in the real world. On the other hand, simulation provides a cheaper and safer way to obtain a vast amount of trial data which is more suitable for RL training. Yet, due to simulation inaccuracies, disparities between simulation and reality inevitably exist and simulated-trained policies transferring directly to the real environment often result in poor performance, which is also known as 'reality gap'.

To bridge the gap between simulation and reality, many studies focus on sim-to-real transfers to translate simulated-trained policy to reality. As one of the most popular sim-to-real methods, Domain Randomization (DR) [3–6] injects random noise into observations, actions, and physical parameters during simulated training, which helps agents adapt to different environments, making the policies

---

[*]Corresponding author

38th Conference on Neural Information Processing Systems (NeurIPS 2024).

more robust. However, DR requires a good understanding of the specific domain to determine the appropriate ranges for these random variations. If these ranges are set inappropriately, it can hinder the successful transfer of policies. Additionally, DR often needs longer training time to ensure adaptability of the policy to various environments, and always results in suboptimal and conservative performance[6]. To tackle these challenges, recent attempts have been made to use real-world data for fostering policy transfer to real-world applications. One approach is optimizing simulator parameters to make the simulation more realistic [7–14]. Unfortunately, these methods still face challenges, such as the need to iteratively collect large amounts of high-cost real-world data, difficulty in applying to high-dimensional control tasks, and instability in the parameter optimization process.

In this work, we introduce Evolutionary Adversarial Simulator Identification (EASI), a novel sim-to-real method that frames sim-to-real transfer as a search problem and employs a unique collaboration between Generative Adversarial Network (GAN) and Evolutionary Strategy (ES). EASI searches for physical parameters to align state transitions between simulation and reality as closely as possible, ensuring that policies trained in simulation seamlessly transfer to the real world. Within EASI, ES acts as a generator in an adversarial competition with a neural network discriminator, which distinguishes state transitions originating from either simulation or reality. The discriminator assigns higher scores to state transitions more likely sampled from reality, acting as the fitness function for ES and guiding the evolution of physical parameter distributions. Concurrently, the discriminator's ability to differentiate between simulated and real state transitions improves through iterative training. EASI features simplicity, low cost, and high fidelity, requiring only a small amount of real-world data to bridge the gap between simulation and reality.

In the experiments, we test EASI on 4 sim-to-sim tasks (Go2, Cartpole, Ant, Ballbalance) and 2 sim-to-real tasks (Go2, Cartpole). We first use Uniform Domain Randomization (UDR) to train a rough initial policy over a wide parameter range, allowing the agent to collect data in the target environment controlled by the initial policy. Then, we employ EASI for simulator identification that makes the simulator most similar to the target domain. The final policy is trained based on these optimized simulator parameters. Our experiments demonstrate that EASI can stably and rapidly search for simulator parameters, and the policies trained within the optimized parameter distribution can effectively transfer to the target domain. Video and code are shown at our page. In particular, we make three main contributions in this work:

1. We propose a novel method, EASI, which employs a unique collaboration between GAN and ES to address sim-to-real challenges.

2. We model sim-to-real transfer as a search problem, searching for simulator parameters in an adversarial process to make state transitions between simulation and reality most similar.

3. We demonstrate the performance of EASI in both sim-to-sim and sim-to-real tasks, showing superior performance compared to existing sim-to-real algorithms.

## 2 Related Work

Transferring simulated-trained policies to the real world in a stable and cost-effective manner has long been a goal for sim-to-real transfers. Previous research has approached sim-to-real transfer from both zero-shot and few-shot perspectives.Zero-shot sim-to-real transfer directly trains a policy capable of operating in reality without using any real-world data. Domain Randomization (DR) is one of the most intuitive and widely applied zero-shot sim-to-real methods[3–6]. DR injects random noise into state observations, actions, and physical parameters to enhance the robustness of the policy, allowing the agent to adapt to various environments. However, DR requires specific domain prior knowledge and hand-engineering to determine the distribution of injected random noise, while also suffering from reduced control performance and high training costs. To improve DR, Akkaya et al.[15] used task performance as a metric to develop a curriculum that gradually increases the level of randomization. Mehta et al.[16] employed GAN to identify the most informative parts of the randomization parameter space, focusing more attention on these areas. Furthermore, some zero shot methods [17–19] introduced an adaptation module or estimator to predict robot state latent variables or explicitly estimate privileged information that the robot cannot access in reality, allowing the robot to adapt to different reality environments relying only on limited proprioception.

Few-shot sim-to-real transfer leverages real-world data to facilitate policy transfer. Domain adaptation is one of the approaches that allows policy to generalize to reality. Image adaptation methods [20–23]

used GAN to transform simulated RGB images in a way that is matched to the reality. Grounded Action Transformation (GAT) [24–26] suggested action transformation that applying the transformed actions in simulation has the same effects as applying the original actions had on reality. In robotic tasks, domain adaptation methods often require a substantial amount of costly real-world data for effective transfer. Inspired by classic system identification [27–29], some efforts focused on optimizing simulators so that the policies trained within them will better adapt to reality. Bayesian optimization was used to find the posterior distribution of simulator parameters based on real-world observations [13, 14, 30]. Chebotar et al.[8] and Chang et al.[31] treated the trajectory mismatch as the cost function and iteratively updated the simulator parameters. Memmel et al. [11] and Marija et al.[32] designed effective real-world exploration policies to sample more informative trajectories for system identification. TuneNet [7] employed supervised learning to train a neural network mapping simulation trajectories to parameter gradients toward the real world. Some approaches [9, 33] used RL to find proper parameters, treating trajectories as RL states and parameters as RL actions. SimGAN [10] employed GAN to distinguish between source and target domain dynamics and enhance the hybrid simulator to make it more realistic. However, most current simulation parameter optimization methods still suffer from instability, high cost of reality data collecting and are only suitable for low-dimensional control tasks.

In this work, EASI employs GAN to differentiate between state transitions in simulation and reality, using ES to search for parameters that align the simulation more closely with the real world based on the GAN's distinction results. EASI requires lower demands on real-world data while enabling faster and more stable searching of simulator parameters. This helps us obtain high-fidelity, low-cost simulators to provide a real world like environment for RL training.

## 3 Preliminary

### 3.1 Domain Randomization

Consider a RL problem defined by the Markov Decision Process (MDP) represented by a tuple $\langle \mathcal{S}, \mathcal{A}, \mathcal{R}, \gamma, \mathcal{P}(\xi) \rangle$, where $\mathcal{S}$ is the state space, $\mathcal{A}$ is the action space, $\mathcal{R} : \mathcal{S} \times \mathcal{A} \to \mathbb{R}$ represents the reward function, $\gamma$ is the discount factor, $\mathcal{P}(\xi) : \mathcal{S} \times \mathcal{A} \times \mathcal{S} \to \mathbb{R}_+$ represents the system's state-transition probability function under simulator parameters $\xi$. The state transition varies depending on the type of system and its specific parameters. We denote the real-world state transition as $\mathcal{P}_r(\xi_{real})$ and the simulation state transition as $\mathcal{P}_s(\xi_{sim})$. In RL process, an agent interacts with the environment and learns a policy $\pi(\mathbf{a}|\mathbf{s})$ to maximize the expected discounted return $\mathbb{E}_{p(\tau|\pi)} \left[ \sum_{t=0}^{T-1} \gamma^t \mathcal{R}(\mathbf{s}_t, \mathbf{a}_t) \right]$, where $p(\tau|\pi) = p(\mathbf{s}_0) \prod_{t=0}^{T-1} p(\mathbf{s}_{t+1}|\mathbf{s}_t, \mathbf{a}_t) \pi(\mathbf{a}_t|\mathbf{s}_t)$ represents the likelihood of a trajectory $\tau = \{\mathbf{s}_0, \mathbf{a}_0, r_0, \mathbf{s}_1, \ldots, \mathbf{s}_{T-1}, \mathbf{a}_{T-1}, r_{T-1}, \mathbf{s}_T\}$ under $\pi$. Due to the existence of the reality gap, policies trained in simulation often perform poorly when directly transferred to the real environment, and the manifestation of the reality gap in MDP lies in the difference between $\mathcal{P}_r(\xi_{real})$ and $\mathcal{P}_s(\xi_{sim})$. To address this issue, DR samples $\xi_{sim} \sim \Xi \in \mathbb{R}^N$ and sets the simulator's parameters to $\xi_{sim}$ at each simulation training round, allowing the RL-trained policy to adapt to various $\mathcal{P}_s(\xi_{sim})$, thus improving system robustness. However, to train policies that perform well in the real world using DR, it is necessary to ensure that there exists $\xi^* \in \Xi$ such that $\mathcal{P}_r(\xi_{real}) \approx \mathcal{P}_s(\xi^*)$, which means it needs specific domain knowledge to determine $\Xi$.

### 3.2 Evolutionary Strategy

Evolutionary Strategy (ES) is a method that mimics biological evolution to solve high-dimensional, continuous-value domain problems. ES algorithm addresses the following search problem: maximize a nonlinear fitness function that is a mapping from search space $\mathbb{R}^d$ to fitness value $\mathbb{R}$. In each iteration (generation), the individual from the previous generation undergoes recombination and mutation to reproduce the next generation of individuals. Each offspring is evaluated through the fitness function, and individuals with the higher scores form the next generation's population.

In this work, we use the discriminator as the fitness function of ES to evaluate the similarity of state transition, guiding ES to find $\xi_{sim}$ that deceives the discriminator to minimize the distance between $\mathcal{P}_r(\xi_{real})$ and $\mathcal{P}_s(\xi_{sim})$.

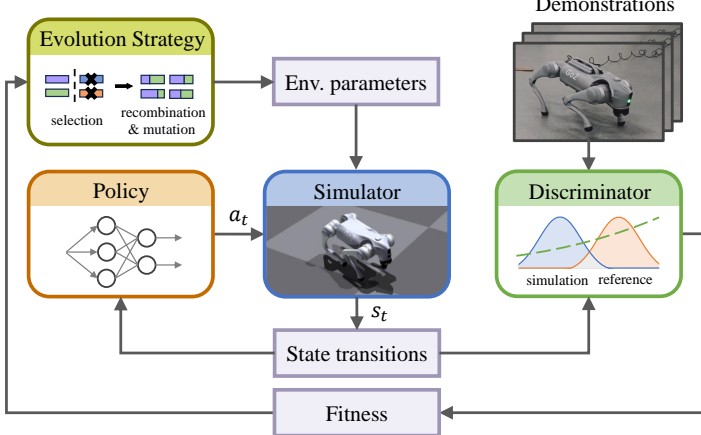

Figure 1: Schematic overview of EASI. ES acts as a generator in adversarial competition with a neural network discriminator distinguishing between simulation and reality state transitions. The discriminator serves as the fitness function, guiding the evolution of the physical parameter distributions.

### 3.3 Generative Adversarial Network

GAN [34] consists two neural networks, generator $G$ and discriminator $D$, engaged in an adversarial process to generate new data with the same statistics as the training set. $G$ and $D$ play a two-player minimax game with value function $V(G, D)$, which can be described as

$$\min_G \max_D V(G, D) = \mathbb{E}_{x \sim p_{data}(x)}[log D(x)] + \mathbb{E}_{z \sim p(z)}[log(1 - D(G(z)))], \tag{1}$$

where $p(z)$ is the prior on input noise variables, $p_{data}$ is the training set. $G$ and $D$ continuously adjust parameters, with the ultimate goal of training a generator strong enough to let the discriminator unable to distinguish whether the outputs of the generator network are generated or sampled from training set.

## 4 Evolutionary Adversarial Simulator Identification

### 4.1 Imitate State Transitions

Modeling the processes in reality and simulation as MDPs, the primary difference between the two MDPs lies in their state transitions $\mathcal{P}_r(\xi_{real})$ and $\mathcal{P}_s(\xi_{sim})$. In this paper, we focus on minimizing the state transition gap between simulation and reality. For simulation with variable physical parameters, we aim to find the distribution of the parameters to achieve

$$\Xi = \arg\min_{\xi_{sim} \in \Xi} \|\mathcal{P}_r(\xi_{real}), \mathcal{P}_s(\xi_{sim})\|. \tag{2}$$

In EASI, we use a discriminator to distinguish whether state transitions come from simulation or reality, the discriminator $D(\mathbf{s}, \mathbf{a}, \mathbf{s}')$ can be trained using

$$\max_D \mathbb{E}_{d^{\mathcal{M}}(\mathbf{s},\mathbf{a},\mathbf{s}')}[log(D(\mathbf{s}, \mathbf{a}, \mathbf{s}'))] + \mathbb{E}_{d^{\mathcal{B}}(\mathbf{s},\mathbf{a},\mathbf{s}')}[log(1 - D(\mathbf{s}, \mathbf{a}, \mathbf{s}'))], \tag{3}$$

where $d^{\mathcal{M}}(\mathbf{s}, \mathbf{a}, \mathbf{s}')$ and $d^{\mathcal{B}}(\mathbf{s}, \mathbf{a}, \mathbf{s}')$ denote the distribution of action $\mathbf{a}$ causing state transition from state $\mathbf{s}$ to $\mathbf{s}'$ in the demonstration dataset and in simulation respectively. As demonstrated in previous work [34], the discriminator serves as a measure of the Jensen-Shannon divergence between $\mathcal{P}_r(\xi_{real})$ and $\mathcal{P}_s(\xi_{sim})$.

It is important to acknowledge that, due to inherent design factors in the simulator, achieving state transitions that are precisely identical to those in the real world may be impossible. Nonetheless, our experiments have shown that EASI significantly enhances the simulator's realism, allowing policies trained in simulation to transfer more effectively to real world.

**Algorithm 1:** EASI

---

**Input:** Initial parameter distribution $\Xi^{(0)}$, motion control policy $\pi_0$, demonstration state transition $\mathcal{M}$, simulation state transition buffer $\mathcal{B}$

**Output:** Parameter distribution $\Xi^*$

**1 for** *generation* $i = 0, 1, 2, \cdots, G$ **do**

**2**     Sample individuals by $\xi_j^{(i)} \sim \Xi^{(i)}$    $j = 1, 2, 3, \cdots, N$

**3**     **for** *simulation environment* $j = 1, 2, 3, \cdots, N$ **do**

**4**        Set simulator parameters to $\xi_j^{(i)}$

**5**        Use $\pi_0$ to sample trajectory $\tau_j$

**6**        Store $\tau_j$ in $\mathcal{B}$

**7**     **end**

**8**     **for** *update step* $= 1, 2, 3, \cdots, n$ **do**

**9**        $b^{\mathcal{M}}$ = sample batch of K state transitions$\{(\mathbf{s}_k, \mathbf{a}_k, \mathbf{s}_k')\}_{k=1}^K$ from $\mathcal{M}$

**10**       $b^{\mathcal{B}}$ = sample batch of K state transitions$\{(\mathbf{s}_k, \mathbf{a}_k, \mathbf{s}_k')\}_{k=1}^K$ from $\mathcal{B}$

**11**       Update $D$ according to Equation 4 using $b^{\mathcal{M}}$ and $b^{\mathcal{B}}$

**12**       Clip network weights of $D$

**13**     **end**

**14**     Calculate discriminator reward $r_j^{(i)} = \mathbb{E}_{\tau_j}[D(\mathbf{s}, \mathbf{a}, \mathbf{s}')]$    $j = 1, 2, 3, \cdots, N$

**15**     Using ES to find next generation distribution $\Xi^{(i+1)} = \text{ES}\left(\xi^{(i)}, r^{(i)}\right)$

**16 end**

---

### 4.2 Wasserstein Loss

The standard GAN discriminator objective detailed in Equation 3 typically uses a binary cross-entropy loss function which tends to encounter the problem of vanishing gradients due to saturation regions [35]. Specifically, if the discriminator is overly powerful, it may excessively classify state transitions, leading to identical bad rewards for all transitions sampled from the simulation. This makes it difficult to identify which parameters yield more realistic state transitions. Additionally, due to inherent discrepancies between the simulator and reality, the state transitions generated by the simulator may not perfectly align with those in the real world, further complicating the challenges posed by a strong discriminator. To address such challenges, WGAN-style discriminator [36] is proposed to mitigate the issue of gradient vanishing and has proven effective for robot-related tasks [2]. In our work, we employ Wasserstein loss to update the discriminator using

$$\max_D \mathbb{E}_{d^{\mathcal{M}}(\mathbf{s}, \mathbf{a}, \mathbf{s}')}[D(\mathbf{s}, \mathbf{a}, \mathbf{s}')] - \mathbb{E}_{d^{\mathcal{B}}(\mathbf{s}, \mathbf{a}, \mathbf{s}')}[D(\mathbf{s}, \mathbf{a}, \mathbf{s}')]. \tag{4}$$

This WGAN-style discriminators is an efficient approximation to the Earth-Mover distance which more effectively and smoothly measures the distance between two probability distributions. We clip the network weights to $[-0.01, 0.01]$ as the original WGAN did to ensure the Lipschitz continuity of $D(\mathbf{s}, \mathbf{a}, \mathbf{s}')$ in the training process. Using Wasserstein loss, we could better evaluate the similarity between simulation and real world state transitions.

### 4.3 Evolutionary Strategy for Parameter Search

Our goal is to find a parameter distribution (e.g. Gaussian distribution) that makes the simulator most similar to the demonstration. The distance between $P_r(\xi_{real})$ and $P_s(\xi_{sim})$ is measured by $D(\mathbf{s}, \mathbf{a}, \mathbf{s}')$ and what we need to do is to find the best parameter distribution $\Xi^*$ that get the highest reward of $D(\mathbf{s}, \mathbf{a}, \mathbf{s}')$, which could be described as

$$\Xi^* = \arg\min_\Xi \max_D \mathbb{E}_{d^{\mathcal{M}}(\mathbf{s}, \mathbf{a}, \mathbf{s}')}[D(\mathbf{s}, \mathbf{a}, \mathbf{s}')] - \mathbb{E}_{d^{\mathcal{B}}(\mathbf{s}, \mathbf{a}, \mathbf{s}')}[D(\mathbf{s}, \mathbf{a}, \mathbf{s}')]. \tag{5}$$

In EASI, we employ the ES as the generator, with the goal of creating a parameter distribution for the simulator that closely mimics the state transitions observed in the demonstration data. EASI uses the ES to select parameters that receive higher rewards from the discriminator, designating these parameters as elites. The elites then undergo recombination and mutation to produce the next

generation of parameters. The discriminator evaluates whether the state transitions in the generated trajectories are sufficiently similar to those of the expert demonstrations, assigning scores based on the degree of similarity. Simultaneously, the discriminator is trained using the trajectories collected during simulation and demonstration, guided by Equation 4. With the selection process driven by the ES, the state transitions produced by the simulator increasingly resemble those in the demonstration data, and the discriminator can better distinguish the gap between simulated and demonstration state transitions. Ultimately, this algorithm yields a set of simulator parameters that align the state transitions in the simulated environment with those in the real world. This optimized parameter set is then utilized to train policies that can be directly applied in real world. The schematic overview of the EASI architecture is shown in Fig. 1, and the pseudo-code is shown in Algorithm 1.

It's notable that when sampling trajectory from simulation and real world, the motion control policy we use does not necessarily match the final desired policy and the robot may be controlled by MPC[37], PID[38], RL, or even manually operated by humans, as long as it can collect state transition containing enough state transition information we need. This characteristic of our algorithm significantly reduces the cost of collecting real data. With our algorithm, using a rough policy run in the real world to collect suboptimal trajectories is enough to find a proper parameter for the simulator.

## 5 Experiment

In this section, we conduct experiments in various tasks to validate the performance of EASI and answer three questions:

1. Can EASI enhance the simulator's similarity to real-world?
2. Can EASI improve performance in sim-to-real transfer tasks?
3. How much real-world data is required for EASI?

**Task Setup**    We test EASI in 4 sim-to-sim tasks illustrated in Fig. 2 and 2 sim-to-real tasks presented in Fig. 3. **(1) Cartpole** environment includes an inverse pendulum connected to a 1-DoF cart. The goal of Cartpole task is to keep the pendulum on the cart balanced for as long as possible. **(2) Go2** environment is a 12-DoF Unitree Go2 quadruped robot, with each leg capable of 3-DoF. The goal of Go2 task is to keep the quadruped run forward and track a specified speed in a natural locomotion. **(3) Ant** environment is an 8-DOF quadruped robot consisting of four legs attached to a common base. The goal of Ant task is controlling the ant run as fast as possible. **(4) Ballbalance** environment consists a table with three legs and a ball placed on the table. Each leg has 1 DoF to control the target angle of the leg joint. The goal of Ballbalance is to stabilize the ball at a target position on the table.

We utilize Isaac Gym [39] as the simulator. In the simulator identification process, we create 300 parallel environments for EASI to evaluate the similarity between reality and simulator with different

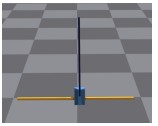 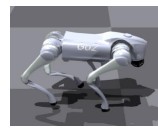 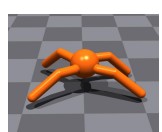 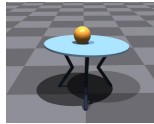

Figure 2: Experiment tasks in simulation: Cartpole, Go2, Ant, and Ballbalance, presented in order from left to right.

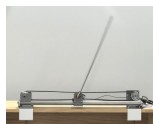 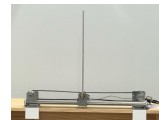 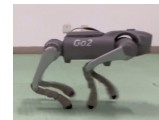 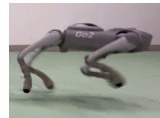

Figure 3: *Left*: Cartpole task in reality, aiming to keep the pendulum on the cart balanced for as long as possible. *Right*: Go2 task in reality, aiming to keep running forward and track a specified speed.

parameters. For Cartpole, Ant, and Ballbalance, we use SAC [40] to train a policy (4 layer MLP) as the locomotion controller. As for Go2, we employ Ess-InfoGAIL [41] to train a quadruped controller.

**Baselines** **(1) Uniform Domain Randomization (UDR)**: Uniformly sampling parameters from $\Xi_{UDR}$ at the beginning of each training iteration. **(2) Oracle**: Directly training in the pseudo-real environment in sim-to-sim experiments, representing the ideal upper bound for sim-to-sim transfer tasks. **(3) FineTune**: Fine-tune the UDR policy using real-world data, enabling the UDR to adapt to the real world in advance to improve transfer performance. **(4) GARAT**: GARAT[26] adjusts actions from the policy such that, after adjustment, the actions applied in the simulator result in state transitions more similar to those in the real demonstration.

**EASI Implementation** There are no specific requirements for the selection of the ES in EASI, mainstream ES such as CMA-ES[42], NES[43] etc. can be used. In this paper, we employ $(\mu/\mu_I, \lambda)$-ES [44] as the generator with the setting $\mu = 150$ and $\lambda = 300$, which means every generation has 300 individuals and 150 elites are chosen to undergo recombination and mutation to generate the next generation. For the collection of the demonstration dataset, we use UDR to train a rough policy, and then use the policy to control agents in the target domain collecting trajectories.

EASI can quickly accomplish parameter searching tasks. The discriminator can directly evaluate the similarity between simulator and reality by leveraging a single trajectory sampled from the simulator with a specific parameter. Furthermore, in Isaac Gym, we can parallelly collect trajectories in hundreds of environments with different parameters which means we could evaluate hundreds of parameters parallelly. In our experiment, running on a PC equipped with Intel i5-13600KF and RTX 4060 Ti, EASI completed the evolutionary adversarial searching process in **less than 10 minutes**.

## 5.1 Sim-to-Sim Policy Transfer

To validate the theoretical capability of EASI for simulator identification, we conduct sim-to-sim experiments in 4 tasks (Cartpole, Go2, Ant, and Ballbalance). Initially, a set of parameters $\xi_r$ is selected as the target parameters for the simulator, which is regarded as the pseudo-real environment. For UDR, we use $\Xi_{UDR} = U[\frac{1}{k} \times \xi_r, k \times \xi_r]$ as the uniform parameter distribution and train an initial policy $\pi_0$ with UDR. Depending on the difficulty of the task, we set $k = 3$ for the Cartpole, Ant, and Ballbalance tasks, and $k = 2$ for Go2 task. Then $\pi_0$ is used to collect 200 trajectories in the pseudo-real environment as demonstration state transition $\mathcal{M}$. During the EASI parameter optimization process, two approaches are used for selecting the initial parameter distribution: Within Distribution (WD) and Out of Distribution (OOD). In the WD experiment, the initial parameter distribution $\Xi^{(0)}$ is consistent with the domain randomization distribution $\Xi_{UDR}$, and the target parameter $\xi_r$ is included within the initial parameter distribution $\Xi^{(0)}$. In the

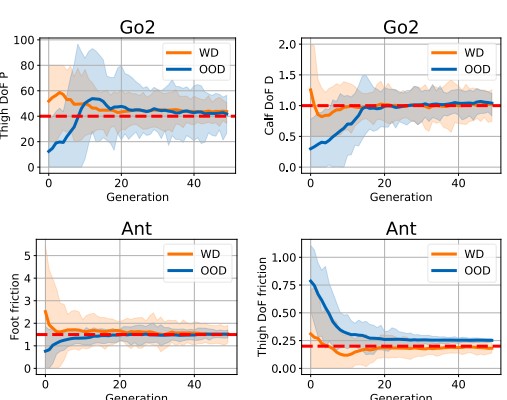

Figure 4: Evolution process of parameters. The red dashed line represents the target parameter. WD means target parameter within initial parameter distribution, and OOD means target parameter out of initial parameter distribution.

OOD experiment, we deliberately set incorrect initial parameter distribution $\Xi^{(0)}$ (e.g. set as $U[0, \xi_r]$ or $U[\xi_r, 2 \times \xi_r]$) so that the target parameter is outside the initial parameter distribution. Taking Go2 and Ant environment as examples, there are 8 parameters for Go2 and 11 parameters for Ant to identify. Here we choose 2 of the parameters in each task to demonstrate the generating process of the parameter. Fig. 4 represents the convergence process of parameters, where the x-axis represents the generations of ES, and the y-axis represents the parameter values. It can be observed from the figure that at the beginning, parameters are uniformly distributed within $\Xi^{(0)}$. As the parameter evolution progresses, the parameter distribution quickly adjusts to the vicinity of the target parameters. This experiment showcases EASI's capability to adjust parameter distributions, even when the initial

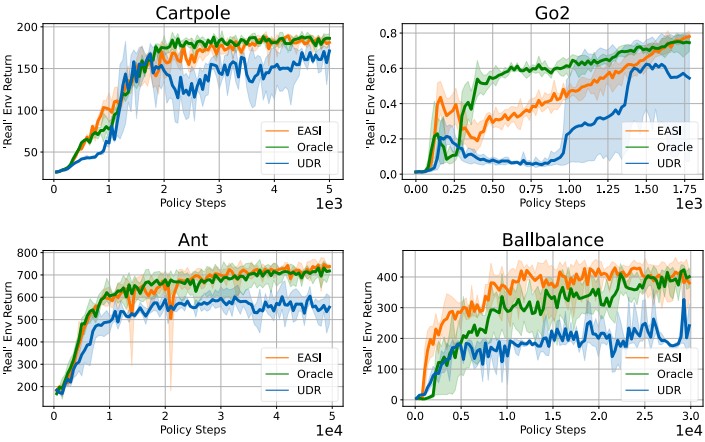

Figure 5: Performance on the pseudo-real environment over the process of training.

distribution is incorrect. This implies that by specifying a rough initial parameter range, EASI can effectively align the parameter distribution within the desired range.

After parameter optimization, we utilize the converged parameter distribution $\Xi^*$ to train new policies. In each task, EASI runs three times with different random seeds to avoid randomness. Figure 5 depicts the policy's performance in the pseudo-real environment throughout the training process. The X-axis represents the number of optimization steps for the policy during training in the simulation, and the Y-axis indicates the policy's performance when tested in the pseudo-real environment. In all tasks, our method outperforms UDR and even demonstrates comparable performance to the policy trained directly in the pseudo-real environment.

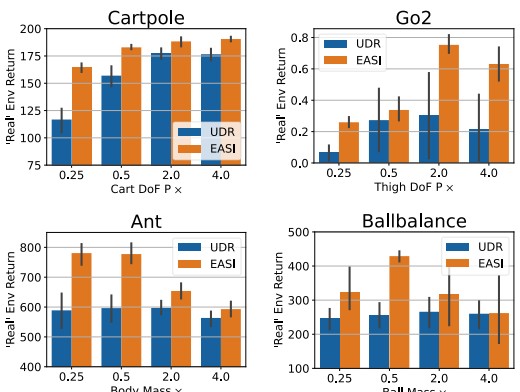

Figure 6: Policy performance in target environments with different parameters.

Subsequently, we conduct a test to evaluate the performance of EASI across various target environments. We fix the initial parameter distribution $\Xi^{(0)} = U[\frac{1}{k} \times \xi_r, k \times \xi_r]$ and adjust one of the parameters $\xi_r^i$ to 1/4, 1/2, 2, and 4 times its original value, denoted as $\xi_r'$. We choose the proportional parameter of the PD controller for the Cart DoF in Cartpole, the proportional parameter of the PD controller for the Thigh Dof in Go2, the body mass of Ant, and ball mass of Ballbalance as the parameters to be adjusted. We use the simulation with parameter $\xi_r'$ as the target environment and utilize EASI to identify. The performance of trained policies in different target environments are shown in Fig.6. It can be seen that the policies trained with UDR are overly conservative or fail to adapt to the target environment due to the gap between source and target environments. On the contrary, simulators optimized by EASI are more similar to the target environment, thus policies trained

Table 1: Policy performance in target environments with different parameters.

| Target Environment | UDR | FineTune | GARAT | EASI |
|---|---|---|---|---|
| CartPole Cart DoF P × 0.25 | 116.8±13.0 | 135.1±7.8 | 144.4±1.2 | **164.4±4.6** |
| CartPole Cart DoF P × 0.5 | 157.0±10.5 | 171.2±8.4 | 161.6±4.5 | **182.9±2.2** |
| Ant Body Mass × 0.25 | 588.9±65.9 | 590.2±18.6 | 327.7±15.9 | **780.7±41.5** |
| Ant Body Mass × 0.5 | 596.4±52.3 | 640.8±30.0 | 228.4±19.2 | **777.4±37.5** |

in these simulations are more adaptable to target environments and achieve higher performance in the target domain. In specific scenarios, we compare EASI with other sim-to-real methods, with detailed results presented in Table 1. The experimental results demonstrate that policies trained with EASI consistently achieve better performance in the target environment, highlighting EASI's effectiveness in facilitating a seamless transfer of policies to the target environment.

## 5.2 Sim-to-Real Policy Transfer

**Cartpole**   In real-world control, using Cartpole as an example, we run the policy on a PC receiving state information from robot and sending action commands to the robot. The control frequency for the Cartpole task is 50Hz. We evaluate the performance of the policy by the angle error of pole and the velocity of cart, which reflect the accuracy and stability of the control policy.

Table 2: Policy performance in reality

| Algotirhm | Angle Error$\times 10^{-2}$ | Cart Vel $\times 10^{-1}$ |
|-----------|------------------------|------------------------|
| UDR | 2.65±0.19 | 1.63±0.03 |
| EASI | 1.67±0.16 | 1.21±0.08 |

We train an initial policy using UDR and sample 50 trajectories including approximately 10000 state transitions in real-world environment. For EASI, initial parameter distribution and other algorithm settings remain the same as sim-to-sim tasks. Table 2 presents the performance of different algorithms in real-world Cartpole tasks, and it can be seen that the policies trained in the simulator optimized by EASI achieve more accurate and stable control in the Cartpole task and show significant improvements compared to previous sim-to-real method.

**Go2**   To further evaluate the performance of EASI in real-world scenarios, we conduct an experiment using the Unitree Go2 quadruped robot. We search for 7 parameters: the PD parameters for the hip, thigh, and calf joints, as well as the body mass. Unlike the sim-to-sim experiment, we do not include ground friction in our search to allow the robot to adapt to various ground types it might encounter. We train the initial policy using UDR in simulation and use the initial policy to collect real-world data, sampling 2000 steps at a control frequency of 50Hz, which provides about 40 seconds of data. Using these real-world samples, we then employ EASI for parameter search.

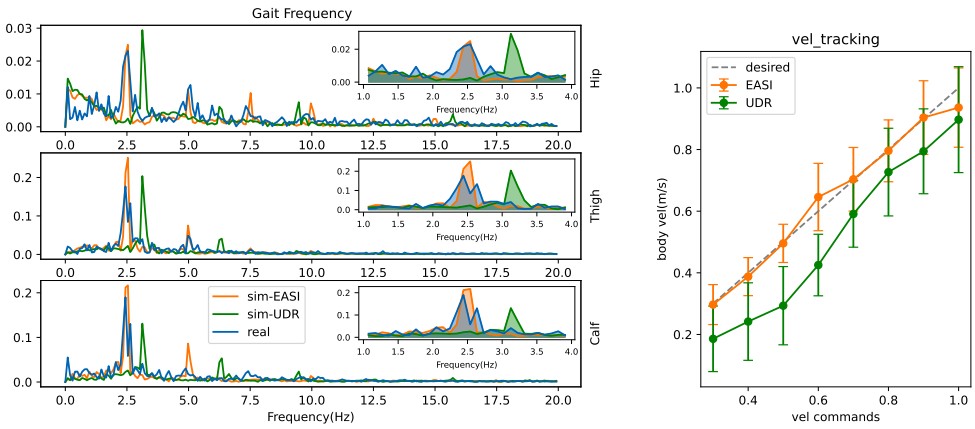

(a) Movement spectrum of the joint.          (b) Velocity tracking performance.

Figure 7: (a) Using the same policy, there is a significant difference in the motion spectrum of the Go2's joints between the simulation and the real environment due to the reality gap. After optimizing with EASI, the motion spectrum of the Go2's joints in the simulation becomes closer to that in the real environment. (b) Comparison of speed tracking in real environments between policies trained with EASI parameter distribution and trained with UDR parameter distribution.

After parameter search with EASI, the simulator become more realistic. In this experiment, we use the same initial policy and speed command (v=1.4m/s) to control the robot's movement in both

simulation and reality environments and plot the frequency spectrum of the robot's joint movements in Fig. 7(a). It is notable that the final output of EASI is the distribution of simulator parameters $\Xi^*$. To elucidate the differences in simulation with optimized parameter distribution $\Xi^*$ and original parameter distribution $\Xi^{(0)}$ (consistent with $\Xi_{UDR}$), we utilize the means of these distributions $\xi^*$ and $\xi^{(0)}$ as the parameters for the simulator, referred to as sim-EASI and sim-UDR, respectively. Despite being controlled by the same policy, there are significant differences in the joint movement frequencies between the sim-UDR and real environments, indicating a substantial reality gap between simulator and the real world. After parameter optimization with EASI, the differences in the joint movement frequency spectrum have been significantly reduced, indicating that the sim-EASI more closely resembles real-world.

Subsequently, the real robot's ability to follow speed commands is tested. Using the parameter distribution $\Xi^*$ obtained from EASI as the simulator parameter distribution to retrain a policy before testing on the real quadruped robot. During the testing phase, we evaluate the robot's performance in the same scenarios using various speed commands. The results in Fig. 7(b) show that policies trained with EASI parameters have better speed tracking capabilities compared to those trained with origin parameters. This also proves that policies trained in the EASI-optimized simulator can perform better in the real world.

### 5.3 Real World Data Requirement

We design a data budget experiment based on sim-to-sim experiments to test the impact of demonstration data quantity on EASI. In this experiment, we limit the number of trajectories sampled in the target domain to 100, 50, and 1 respectively, and conduct EASI using the limited trajectories.

Table 3: Performance in target environments with varying numbers of reference trajectories

| Trajectories | UDR | 1 | 50 | 100 | 200 |
|---|---|---|---|---|---|
| Cartpole | 161.3±15.3 | 185.1±3.1 | 182.7±2.7 | 181.1±4.4 | 182.9±4.3 |
| Ant | 557.4±55.6 | 724.2±7.5 | 747.0±25.0 | 703.1±63.1 | 724.0±31.5 |
| Ballbalance | 226.9±61.7 | 360.1±53.8 | 421.3±26.1 | 428.5±16.9 | 393.1± 60.5 |
| Go2 | 0.62±0.25 | 0.44±0.28 | 0.80±0.06 | 0.72±0.05 | 0.78±0.02 |

The experimental results of the data budget are presented in Table 3. The results demonstrate EASI's data efficiency, as only a small amount of real-world data is needed for EASI to identify parameters that align the simulation closely with the target environment and enhance the policy's adaptability in the real world.

## 6 Limitations

Even though EASI requires relatively low amounts of real-world data, it still necessitates collecting corresponding state transition data for each specific scenario and maintaining certain standards regarding sensor error ranges. When sensors exhibit significant noise, additional techniques for imitation learning from imperfect demonstrations are needed. Furthermore, even within the same scenario, substantial differences in task styles can reduce EASI's performance. To achieve optimal results, it is recommended to collect expert data separately for each task.

## 7 Conclusion

In this work, we introduce EASI, a novel sim-to-real method that combines GAN with ES. EASI uses ES as a generator in an adversarial competition with a neural network discriminator to find physical parameter distributions that align state transitions between simulation and reality. The discriminator serves as the fitness function for ES, guiding the evolution of parameter distributions. We conducted a series of experiments to validate EASI's effectiveness and data efficiency in both sim-to-sim and sim-to-real tasks, demonstrating its exceptional performance. We believe EASI is a general sim-to-real method characterized by simplicity, low cost, and high fidelity. In future work, we will evaluate EASI's performance on a broader range of robotic platforms and industrial scenarios.

## Acknowledgments and Disclosure of Funding

This work was supported in part by the National Natural Science Foundation of China (Nos. 62073160 & 72394363) and the Nanjing University Integrated Research Platform of the Ministry of Education-Top Talents Program.

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

# Supplementary

## A  Full parameter search

To explore the performance of EASI as the number of simulation parameters increases, we conducted a sim-to-sim experiment with the Go2 environment, searching for 25 parameters. In the experiment, we searched for the PD parameters of all motors in the Go2 robot and the mass of the robot. The results of the parameter search are shown in Fig. 8 (Due to space limitations, 10 out of 25 results are displayed).

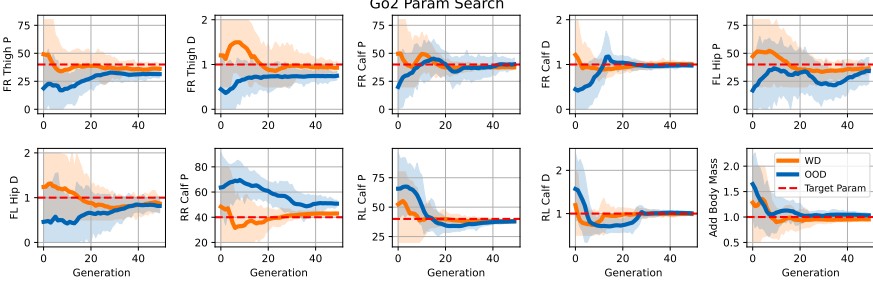

Figure 8: The convergence process for different parameters being searched in the Go2 environment. WD means target parameter within initial parameter distribution, while OOD means out of initial parameter distribution.

## B  Hyperparameter sensitivity analysis

Since EASI uses Evolutionary Strategies (ES) as the simulation parameter generator, it is much more stable in the training process compared to the original GAN algorithm and less sensitive to hyperparameters. In EASI, we select two hyperparameters for sensitivity analysis in Table 4 and Table 5 ,the evaluation metric is the L2 error between the searched parameter and the target parameters.

We first analyze the hyperparameter $\mu/\lambda$. In each evolution, we test $\lambda$ individuals and select the best-performing $\mu$ elite individuals for recombination and mutation. Generally, a higher $\mu/\lambda$ value helps maintain diversity in the population but may slow down the evolution process. Conversely, if $\mu/\lambda$ is too low, a lack of diversity can lead to local optima. To enhance parameter search, we recommend using a larger $\mu/\lambda$ and increasing the number of evolution steps, which will improve EASI's performance but also increase computational time.

Next, we analyze another hyperparameter epoch_disc, which refers to the number of times the discriminator is trained before each generation of evolution. Insufficient training may prevent the discriminator from achieving optimal performance, while excessive training can cause overfitting. In our experiments, we found that varying epoch_disc had only a minor effect on the final search results.

Table 4: Hyperparameter sensitivity analysis for EASI ($\mu/\lambda$).

| $\mu/\lambda$ | 0.1 | 0.3 | 0.5 | 0.7 | 0.9 |
|---|---|---|---|---|---|
| Param Search Error | 1.35±0.03 | 0.63±0.11 | 0.34±0.20 | 1.99±0.13 | 1.44±0.04 |

Table 5: Hyperparameter sensitivity analysis for EASI (epoch_disc).

| epoch_disc | 2 | 5 | 10 | 15 | 20 |
|---|---|---|---|---|---|
| Param Search Error | 0.60±0.08 | 0.19±0.14 | 0.34±0.20 | 0.82±0.06 | 0.46±0.04 |

## C Broader Impacts

In our work, we have proposed EASI for simulation trained strategy transfer to reality. This could potentially drive the application of robotics technology in the real world, thereby positively impacting societal operational efficiency. Sim-to-real techniques reduce the need for extensive physical trials, minimizing costs associated with hardware testing and prototyping. This can accelerate the development of robotic technologies, making them more accessible to a broader range of industries.

However, there is a risk that developers may become overly reliant on simulation results, potentially leading to systems that perform well in simulated environments but fail in the real world due to unmodeled complexities. Additionally, it is important to recognize that the capabilities of simulations have limits, and we cannot expect simulations to perfectly mirror reality. Resources should still be invested in designing better robot structures and developing more advanced simulators.

Moreover, there are growing concerns about the computational resources required for current training schemes. The increasing demand for computational power undoubtedly consumes significant amounts of energy and resources. We are continually exploring more efficient algorithms to address this challenge.

