# OpenReview forum: "EASI: Evolutionary Adversarial Simulator Identification for Sim-to-Real Transfer"
_NeurIPS.cc/2024/Conference — NeurIPS 2024 poster_

### Official Review · Reviewer_yoqm · 2024-06-17

**Soundness:** 3
**Presentation:** 3
**Contribution:** 3
**Rating:** 6
**Confidence:** 3

**Summary:**

The paper introduces an approach to address the challenges of transferring reinforcement learning (RL) policies from simulation to real-world applications. Traditional methods like Domain Randomization (DR) require significant domain-specific knowledge and extensive training time, which often result in suboptimal performance. EASI combines Generative Adversarial Networks (GAN) and Evolutionary Strategies (ES) to identify physical parameter distributions that align state transitions between simulation and reality, thus minimizing the reality gap.

**Strengths:**

#### Originality

- **Approach**: Combining Generative Adversarial Networks (GAN) with Evolutionary Strategies (ES) to address sim-to-real transfer is creative.

#### Quality

- **Low Data Requirement**: The method's effectiveness with minimal real-world data is a significant advancement.

#### Clarity

- **Clear Exposition**: The paper is well-written and explains complex concepts clearly, making it accessible to a broad audience.

#### Significance

- **Practical Impact**: EASI addresses the reality gap in robotics and reinforcement learning, providing a cost-effective and efficient solution with significant practical implications.

- **Broad Applicability**: The method's success in various tasks suggests potential for wide adoption across different domains.

In summary, the paper offers an interesting solution to a critical problem in reinforcement learning.

**Weaknesses:**

#### Methodological Concerns

1. **Wasserstein GAN loss**: The Wasserstein GAN loss was first introduced by the Wasserstein GAN. Using the Wasserstein GAN loss with clipped network weights in the range [-0.01, 0.01] does not seem to be a novel contribution by the author.



#### Experimental Limitations

1. **Diversity of Tasks**: While the paper presents experiments on various tasks, the real-world applications are somewhat limited. For instance, in Sim to Real experiments, only Cartpole is used. Expanding the experiments to include more complex real-world scenarios, such as robot dogs or navigation tasks, would strengthen the claims about EASI's broad applicability.



#### Practical Considerations

1. **Computational Resources**: The computational cost of EASI, especially for training GANs and performing evolutionary searches, is not thoroughly discussed. Providing insights into the computational requirements and potential optimizations would be valuable for practical deployment.

2. **Scalability**: The scalability of EASI to high-dimensional state spaces and action spaces is not well-addressed. Including discussions or preliminary results on scaling the method to more complex systems would be beneficial.

**Questions:**

1. **Parameter Sensitivity and Selection**:

   - **Question**: How sensitive is EASI to the choice of hyperparameters for both the GAN and the ES components?

   - **Suggestion**: Provide a detailed sensitivity analysis or guidelines for selecting these parameters to help with replicability and optimization in different settings.



2. **Diversity of Experimental Tasks**:

   - **Question**: How does EASI perform across more complex real-world scenarios?

   - **Suggestion**: Expand the experimental validation to include other real-world applications (for example, navigation) to strengthen claims about the method’s general applicability.



3. **Computational Resources**:

   - **Question**: What are the computational requirements for training GANs and performing evolutionary searches in EASI?

   - **Suggestion**: Discuss the computational costs and potential optimizations to make the method more accessible for practical deployment.



4. **Scalability**:

   - **Question**: How scalable is EASI to high-dimensional state and action spaces?

   - **Suggestion**: Address the scalability of the method and include preliminary discussions on scaling to more complex systems.

**Limitations:**

The limitations are thoroughly discussed in Section 6.

---

> ### Author Rebuttal · Authors · 2024-08-07
>
> # Response to Reviewer yoqm
>
> `Q1. How sensitive is EASI to the choice of hyperparameters for both the GAN and the ES components?`
>
> `A1.` We conducted a hyperparameter sensitivity analysis for EASI, as detailed in **common response A5**, and provided recommendations for parameter settings. EASI has low sensitivity to hyperparameters, and we rarely changed EASI's hyperparameters across all experiments in this paper. We believe that EASI can achieve plug-and-play effectiveness in other tasks.
>
> `Q2. How does EASI perform across more complex real-world scenarios?`
>
> `A2.` We tested EASI's sim-to-real performance in the real world using the Unitree Go2 robot in **common response A2**. The Go2 task is complex and challenging, yet EASI is able to adjust the simulation parameters using approximately 40 seconds of real demonstration data, making the simulation more realistic.
>
> `Q3. What are the computational requirements for training GANs and performing evolutionary searches in EASI?`
>
> `A3.` EASI's computation consists of three main parts: sim trajectory collection, discriminator training, and parameter evolution.
> 1. Sim Trajectory Collection: This is the most computational-consuming step. Assuming we have $\lambda$ offspring in the evolutionary strategy, we need to use same policy to collect $\lambda$ trajectories from environments with $\lambda$ different parameters. Due to the high parallelism of the Isaacgym simulator, we can generate $\lambda$ environments with different parameters in Isaacgym and sample $\lambda$ trajectories concurrently.
> 2. Discriminator Training: This involves sampling from both demonstration and environment trajectories to train the discriminator.
> 3. Parameter Evolution: In this part, we use the discriminator to estimate the similarity between the simulation trajectory and the demonstration. Then, using ES to generate the distribution of the next generation of simulation parameters. Based on this new parameter distribution, we sample and obtain the physical parameters of N simulated environments and set the parameters for each environment.
>
> In our experiments with the most complex environment, Go2, setting $\lambda$=300 and trajectory_length=500，we test the time consuming on a PC equipped with Intel i5-13600KF and RTX4060 Ti. For one generation of evolution, the time consuming is as follows:
>
> |  process   | time consuming(s)  |
> |  :------: | :------:  |
> | sim trajectory collection  | 6 |
> | discriminator training  | 0.09 |
> | parameter evolution  | 0.11 |
>
> Typically, 30 to 50 generations are sufficient for parameter convergence.
>
> `Q4. Discuss potential optimizations to make the method more accessible for practical deployment.`
>
> `A4.` Currently, EASI has demonstrated strong performance for real-world deployment. However, we believe there is still significant potential for improvement. Real-world data collection often lacks the ideal conditions found in simulations, and the rough policy trained with UDR may not perform optimally in practical applications. As a result, the demonstration data we gather in real-world settings may be imperfect and unbalanced. As noted in the limitations, there are existing imitation learning methods designed to handle unbalanced and imperfect demonstrations. In the future, we aim to implement similar approaches to train a more effective discriminator based on these imperfect, unbalanced datasets, thereby reducing the deployment costs associated with EASI.
>
> `Q5. How scalable is EASI to high-dimensional state and action spaces?`
>
> `A5.` In the Go2 sim-to-real experiment detailed in **common response A2**, the task featured high-dimensional state and action spaces. Despite these complexities, EASI demonstrated excellent performance in this sim-to-real task.

---

> > ### Comment · Reviewer_yoqm · 2024-08-11
> >
> > Thanks for your rebuttal. My concerns have been resolved and I decide to improve my score.

---

> > > ### Author Response · Authors · 2024-08-12
> > > **Response to Reviewer yoqm**
> > >
> > > Dear Reviewer yoqm:
> > >
> > > Thank you for your detailed questions and suggestions，they have been very enlightening for us. If you have any further questions, please contact us.
> > >
> > > Sincerely, thank you for your time.

---

### Official Review · Reviewer_ZLtE · 2024-07-11

**Soundness:** 2
**Presentation:** 3
**Contribution:** 2
**Rating:** 6
**Confidence:** 4

**Summary:**

This work studies the sim-to-real transfer using an evolution strategies with a learned discriminator. The discriminator learns to distinguish the state transition between real and simulation. The evolution strategies aim to optimize the parameters of the simulator so that the data generated by the simulator is undistinguishable by the discriminator. Experiments on sim-to-sim and sim-to-real cases have been conducted.

**Strengths:**

The proposed EASI framework is easy to understand and reasonable. The three research questions listed in the experiments are important and have been demonstrated.

**Weaknesses:**

The details of the employed GAN architecture lacks furture description.

**Questions:**

1) It seems that the major technical contribution of this work is to learn an objective function for the ES using GAN, for simulator calibration or say parameter inference. Thus, I am very curious about the architecture of the used GAN and how are the tuple s, a, s' processed in the training. More descriptions on the details of used GAN are welcome.
2) In Figure 3, why the EASI can outperform the oracle that indicates the upperbound of the performance?
3) For sim-to-transfer case, the oracle may not represent the upperbound, in which case some advanced compared algorithms need to be compared in the experiments.
4) It is interesting to see EASI can adjust the parameters out of the initial range. What makes it capable of doing so?
5) How is the scalability of the EASI, I mean if the number of parameters increases, what will happen to EASI?

**Limitations:**

YES

---

> ### Author Rebuttal · Authors · 2024-08-07
>
> # Response to Reviewer ZLtE
> `Q1. The architecture of the used GAN and how are the tuple s, a, s' processed in the training. `:
>
> `A1.` The input to the discriminator consists of a state transition $(s, a, s')$. This $(s, a, s')$ data is combined into a tensor with a shape of $(2 \times \text{dim(state)} + \text{dim(action)})$, which is then fed into a multilayer perceptron (MLP) discriminator. The discriminator is trained using a WGAN-based approach to determine whether the $(s, a, s')$ data originates from the demonstration. Subsequently, we employ Evolution Strategies (ES) to select parameters that make the simulation closely resemble the real environment and to 'generate' the next generation of environment parameters. Unlike conventional GANs, EASI does not utilize a neural network generator; instead, it relies on ES to choose and generate the next set of parameters based on the discriminator's output. Thanks to the stability of ES, EASI achieves faster and more stable training compared to traditional GANs.
>
> In our experiments, an MLP discriminator is sufficient to achieve good results. We also considered employing networks such as LSTM or TCN for processing $(s, a, s')$ data. However, since the MLP has already demonstrated exceptional performance in the current tasks, we plan to explore these alternative network architectures in future work.
>
> `Q2. In Figure 3, why the EASI can outperform the oracle that indicates the upperbound of the performance?`
>
> `A2.` BallBalance is a unique environment where the ball can only be controlled indirectly by a movable table and cannot be directly influenced by actions. In this task, if the ball falls off the table, the episode ends and a penalty is incurred. We hypothesize that during the early stages of ORACLE training, the policy may prioritize maximizing rewards by keeping the ball at the target position, which could inadvertently lead to the ball falling. In contrast, EASI narrows the parameter range but still allows for variations across different environments. As a result, the policy may adopt a more cautious approach in the early training stages, focusing on preventing the ball from falling off and thus avoiding penalties. As training progresses, both ORACLE and EASI can effectively keep the ball from falling while maintaining its position at the target, leading to similar performance in the later stages of training.
>
> On the other hand, UDR, with its overly wide parameter range, leads to an overly conservative policy. This policy only achieves the basic reward of preventing the ball from falling off but is hard to maintain the ball at the target position.
>
> `Q3. For sim-to-transfer case, the oracle may not represent the upperbound, in which case some advanced compared algorithms need to be compared in the experiments. `
>
> `A3.`In the **common response A4**, we introduced additional comparative experiments, including the FineTune and GARAT (a GAT branch method). In these experiments, EASI consistently demonstrated the best performance in transferring simulation trained policies to the target environment.
>
> `Q4. It is interesting to see EASI can adjust the parameters out of the initial range. What makes it capable of doing so?`
>
> `A4.`In EASI, we utilize Evolution Strategies (ES) to adjust the distribution of parameters. During the evolution process, ES incorporates mechanisms such as "recombination" and "mutation."
>
> For recombination: Parameters that contribute to a more realistic environment have a higher probability of producing offspring. As a result, parameters within the initial range that are closer to the target value exert greater influence on the next generation's parameter range, helping to refine the distribution towards the target. For mutation: Each time new offspring are generated, some undergo mutations that deviate from the original distribution. If these mutated individuals achieve higher fitness values than those within the original distribution, the population will gradually evolve in the direction of the mutated individuals. This means that even if the initial parameter range does not include the global optimum, EASI can potentially discover new parameters close to the optimum through multiple iterations.
>
> After refining the simulator's parameter distribution with EASI, we can train or fine-tune the policy in this enhanced simulator, thereby improving sim-to-real performance.
>
> `Q5. How is the scalability of the EASI, I mean if the number of parameters increases, what will happen to EASI?`
>
> `A5.` In **common response A3**, we introduced additional simulation parameters, searching for a total of 25. EASI was still able to identify the appropriate target parameter range.
>
> In our real-world Go2 sim-to-real experiment discussed in **common response A2**, we only searched across 7 parameters. We set the motors on the four legs to the same positions, resulting in identical parameters for the hip, thigh, and calf motors, each with two PD parameters. Despite this limited search, we achieved excellent sim-to-real performance. By focusing on a few key parameters with EASI, we were able to significantly minimize the gap between the simulation and the real environment.

---

> > ### Comment · Reviewer_ZLtE · 2024-08-09
> > **Official Comment by Reviewer ZLtE**
> >
> > Thank you for the detailed response.
> > Actually, I like the idea of this paper. And all my previous concerns have been addressed.
> > Very appreciated for your works!

---

> > > ### Author Response · Authors · 2024-08-10
> > > **Response to Reviewer ZLtE**
> > >
> > > Dear Reviewer ZLtE:
> > >
> > > We sincerely appreciate your valuable suggestions for our work. We will incorporate some of the content you mentioned in future versions of the paper to make it more complete and solid.
> > >
> > > Sincerely, thank you for your time.

---

> > > ### Author Response · Authors · 2024-08-13
> > >
> > > Dear Reviewer ZLtE:
> > >
> > > Thank you once again for your insightful comments and helpful suggestions. As the deadline for author-reviewer discussions is approaching, if you have any further questions or concerns, please let us know. Thank you very much for your time.

---

> > > > ### Comment · Reviewer_ZLtE · 2024-08-14
> > > > **Official Comment by Reviewer ZLtE**
> > > >
> > > > I also read the rebuttal of the authors with other reviewers. Thanks the authors for the detailed and effective responses. I raise my score to 6.

---

> > > > > ### Author Response · Authors · 2024-08-14
> > > > > **Response to Reviewer ZLtE**
> > > > >
> > > > > Dear Reviewer ZLtE:
> > > > >
> > > > > We greatly appreciate your time in reviewing our paper and reading the follow-up rebuttal! We're thrilled for your recognition of our work. Thank you very much!
> > > > >
> > > > > Best regards,
> > > > >
> > > > > The Authors

---

### Official Review · Reviewer_zVKJ · 2024-07-12

**Soundness:** 3
**Presentation:** 4
**Contribution:** 3
**Rating:** 7
**Confidence:** 5

**Summary:**

This submission presents an approach to sim2real transfer by predicting better simulation hyperparameters via Evolutionary Adversarial Simulator Identification (EASI). EASI is a combination of a generative adversarial network (GAN) and evolutionary strategy approach to generating simulation hyperparameters. Concretely however in the GAN setup, the generator is not another neural network, only the discriminator is a neural network that predicts whether a state transition $(s, a, s')$ is real or not. The generator is now an evolutionary search procedure that generates a distribution of simulation hyperparameters that strives to maximize discriminator rewards. Importantly EASI does not simply generate fixed simulation hyperparameters, but rather a range/set of hyperparameters and can be seen as a more "educated" domain randomization approach when used for training. The result of EASI is better sim2real transfer of policy performance when compared to a standard uniform domain randomization technique. Empirical tests are on robotics control tasks tested can match the oracle.

**Strengths:**

- A very unique approach to sim2real by finding ways to generate a range/set of hyperparameters to try and test on. The approach is very lightweight and is built on top of an initial uniform domain randomization (UDR) + training procedure to then find a better set of simulation hyperparameters.
- Equations and losses are clearly laid out and well-motivated. The algorithm is easy to understand and reason with.
- The performance of EASI looks great. I would hypothesize that UDR fails to do as well as EASI for some tasks because the RL policy spends too much time training on environments with hyperparameters too far away from the real world / don't help with sim2real transfer. It would be good to see additional analysis on why exactly UDR struggles a lot compared to EASI in terms of performance under a fixed online budget.

**Weaknesses:**

- Only control tasks are tested which generally are easier to solve with just domain randomization (albeit EASI does better here). It would be useful to see more complex tasks beyond simple control tasks that leverage manipulation as well. This point doesn't lower my score but if more complex tasks were analyzed I'd be happy to raise the score further to reflect the bigger impact of the method.
- To leverage EASI some sim+real data is required still to train the discriminator and run the evolutionary search. While the amount of real data in experiments can be kept to just one expert demonstration, it's not clear how much simulated data is needed, and what distribution of simulated data is needed. Section 5 mentions UDR is used to train a rough policy but how rough is rough? What if UDR does not get any useful data / what would be considered sufficient UDR to then kickstart EASI?
- Since EASI has to run UDR itself first to get some sim data to train on, overall it could be that running UDR for as long as it takes to run UDR + EASI achieves the same performance, although Figure 3 suggests otherwise. There don't seem to be figures/information about how long the first UDR is done for EASI. It might be fairer to run the UDR baseline for as many online steps EASI uses for training after EASI and before with the rough UDR.
- As pointed out in the limitations section, this sim2real setup requires measuring state in the real world and simulation. For control tasks where the state often is just the robot state and can be measured via sensors, this is fine. For more complex tasks involving manipulation of other objects, this approach will probably not work well.

Happy to raise my score if the above and questions are addressed

**Questions:**

Questions:
- How come in ball balance the oracle method (SAC training on dense rewards with the correct sim parameters) performs worse than EASI, which has SAC training on sim parameters predicted by rough UDR.
- Are dense rewards used for all tasks?
- What if UDR does not get any successful demonstrations e.g. in a cartpole task say the pole is never swung up completely. It sounds like in this scenario EASI might not generate a good set of hyperparameters that would be important for unseen parts of the task?
Typos:
- What does "domain priority" in section mean? The line "However, DR needs specific domain priority and hand-engineer to determine the
79 distribution of injected random noise." might have typos and grammar issues

**Limitations:**

A limitation section is provided and adequately points out the assumption that EASI relies on expert trajectories and accurate state estimation of the same observation data in simulation and the real world.

---

> ### Author Rebuttal · Authors · 2024-08-07
>
> # Response to Reviewer zVKJ
>
> `Q1. How come in ball balance the oracle method performs worse than EASI.`
>
> `A1.` BallBalance is a unique environment where the ball can only be controlled indirectly by a movable table and cannot be directly influenced by actions. In this task, if the ball falls off the table, the episode ends and a penalty is incurred. We hypothesize that during the early stages of ORACLE training, the policy may prioritize maximizing rewards by keeping the ball at the target position, which could inadvertently lead to the ball falling. In contrast, EASI narrows the parameter range but still allows for variations across different environments. As a result, the policy may adopt a more cautious approach in the early training stages, focusing on preventing the ball from falling off and thus avoiding penalties. As training progresses, both ORACLE and EASI can effectively keep the ball from falling while maintaining its position at the target, leading to similar performance in the later stages of training.
>
> `Q2. Are dense rewards used for all tasks?`
>
> `A2.` Yes, all experiments were conducted using dense rewards during training. EASI mainly focuses on the state transitions of these tasks, and the training method of the task itself is not the primary focus of EASI.
>
> `Q3. What if UDR does not get any successful demonstrations? `
>
> `A3.` For EASI, our requirement for realworld demonstration is that the $(s,a,s')$ tuples contain sufficient information, meaning that the state transition distribution in the demonstration needs to partly overlap with that in the simulation. If the rough UDR policy can run in the real environment, even if the performance is not ideal, it can still provide valuable state transitions for EASI training.
>
> If UDR completely fails to accomplish the task, but some meaningful $(s,a,s')$ tuples are still recorded during action execution. In this situation, the collected demonstration data may be imperfect and unbalanced, with only a small portion of state transition overlap with that in the simulation. In such cases, EASI may fail due to insufficient state transition information.
> We mentioned this limitation in the paper, and notice that current imitation learning methods address imperfect and unbalanced demonstrations. In the future, we will aim to use imperfect and unbalanced imitation learning methods to extract useful state transition information from failed real-world demonstrations and then use EASI to find suitable parameters. We believe that in future work, EASI could have lower requirements for real demonstrations, allowing it to identify optimal parameters even when the policy complete very poorly in the reality.
>
>
> `Q4. What does "domain priority" in section mean? `
>
> `A4.` What we meant to express is that Domain Randomization (DR) requires prior knowledge of the specific domain, including detailed parameters about real-world deployment. Thank you for pointing out the issues with the wording in the paper, and we will revise the wording to ensure clearer understanding.
>
>
>
> `Q5. Only control tasks are tested which generally are easier to solve with just domain randomization.`
>
> `A5.` The gap between simulation and real-world environments has long been a significant challenge for deploying RL-based robotic controllers in practical applications. Through our method EASI, we aim to advance the deployment of such controllers, focusing specifically on control tasks. And additionally, we tested EASI in a more complex sim-to-real tasks using the Unitree Go2 robot in **common response A2**.
>
> `Q6. it's not clear how much simulated data is needed.`
>
> `A6.` During EASI's process, $\lambda$ environments with different parameters are created in the Isaac Gym simulator (in the experiment, we set $\lambda=300$), and use a same policy to collect 1 trajectory for each sim environment, the max_trajectory length is setted in different tasks. Thanks to Isaac Gym's high degree of parallelism, we can parallelly run hundreds or even thousands of environments with different physical parameters. After each evolutionary iteration, all environments are reconfigured with new physical parameters based on the evolutionary strategy, initiating the next evolution cycle. The process of generating sim data is highly convenient, fast, and inexpensive, so the amount of simulation data used is not a major concern.
>
> `Q7. It might be fairer to run the UDR baseline for as many online steps EASI uses for training after EASI and before with the rough UDR.`
>
> `A7.` To ensure fairness, we used the same real demonstration data as EASI to fine-tune the UDR algorithm. The specific experimental details are described in **common response A4**. In more complex environments like Ant, FineTune's performance improvement is limited when the amount of real demonstration data is insufficient. EASI, on the other hand, consistently achieves significant improvements in policy transfer to target environment.
>
> `Q8. For more complex tasks involving manipulation of other objects, this approach will probably not work well.`
>
> `A8.` For complex tasks involving manipulated objects, if the parameters of the object are known, our ball balance experiment can be seen as such a task, where a 'manipulated ball' is placed on the table. If the parameters of the manipulated object are uncertain, we can still utilize EASI to adjust the manipulator's parameters, allowing the simulation to more closely resemble the target environment while accommodating a wider range of parameters for the manipulated object.

---

> > ### Comment · Reviewer_zVKJ · 2024-08-09
> > **Response**
> >
> > Good to see the limitations are mentioned in the paper, I must have missed one or two of them.
> >
> > All concerns except for Q7 are addressed. I think Q8 can't be addressed easily anyway but if somehow EASI can achieve that level of ability with harder manipulation tasks I would say this has a lot more impact than I currently think. Not to say it is not possible but without empirical results I cannot make a fair judgement here.
> >
> > Regarding Q7, I understand that you ensure the same data is fed into EASI and UDR. However there are likely computing differences right? In EASI you are using the EASI setup + running UDR, meaning additional computation and potentially runs very slowly. This does not change my opinion however that EASI is much better. The figures suggest that UDR converges to some suboptimal results compared to EASI. Just pointing out some may think this is an issue and may request wall time results (which is not standard in RL but if you use GPU simulation it really should be since sample efficiency is always poor with high numbers of parallel environments).
> >
> > I will raise my score to 7 as I think this is a technically and well done paper. I choose not to raise higher as there is still a big limitation of having to measure real world states and the benchmarking goes as far as just locomotive tasks (which have already been solved quite well via other methods like straight RL+reward tuning so I find this work less impactful).

---

> > > ### Author Response · Authors · 2024-08-10
> > > **Response to Reviewer zVKJ**
> > >
> > > Dear Reviewer zVKJ:
> > >
> > > We also hope to validate EASI's capabilities in challenging manipulation tasks and we plan to test EASI's performance in the Mujoco Pusher environment—a task where a robotic arm pushes a small ball to a target location—during the Discussion Period, and we will provide you the results as soon as possible.
> > >
> > > For Q7, when using the Isaac Gym GPU simulator running on a RTX 4060Ti, EASI actually runs quite quickly. We measured the time required for EASI in the most complex Go2 environment. In this experiment, we used 300 environments for evolution, meaning that each evolution step involves collecting 300 trajectories in 300 environments with different parameters. In each round of evolution, we measured the time required for each part:
> > >
> > > |  process   | time consuming(s)  |
> > > |  :------: | :------:  |
> > > | sim trajectory collection  | 6 |
> > > | discriminator training  | 0.09 |
> > > | parameter evolution  | 0.11 |
> > >
> > > Generally, parameter convergence can be achieved within 30 to 50 generations. The parameter search with EASI can be completed in less than 10 minutes. Compared to RL training times, which can take several hours or even days, EASI is already very fast.
> > >
> > > While using UDR directly can save the time spent on parameter search, as mentioned in the paper, we often lack the knowledge to set a reasonable parameter range for UDR. Without specialized knowledge to adjust the UDR parameter range, the trained policy might not transfer well to the real environment. Thus, directly using UDR may require extensive expertise and trial and error, which can be time-consuming compared to using EASI.
> > >
> > > Finally, your valuable and insightful feedback has been extremely helpful to us. The issues you raised have greatly help to our work. We sincerely thank you for your time and suggestions!

---

> > > > ### Comment · Reviewer_zVKJ · 2024-08-13
> > > > **Response**
> > > >
> > > > Thanks for the clarification. I'm glad you are considering a pushing task but I still would consider such a task just as easy (if not easier) than locomotive tasks even. It is hard to say as this is somewhat subjective. My bar for challenging manipulation is actual grasping (with gripper or dextrous hands) of at minimum a cube, but better to be e.g. YCB dataset objects.

---

> > > > > ### Author Response · Authors · 2024-08-14
> > > > > **Response to Reviewer zVKJ**
> > > > >
> > > > > Dear Reviewer zVKJ:
> > > > >
> > > > > Thank you for your suggestions. We are currently working hard to validate EASI's performance on more tasks, including pusher (completed), grasping, and dexterous manipulation. However, training to complete all of these tasks requires more time, making it difficult to provide final experimental results during the discussion. We will continue to refine these experiments and plan to include them in a future version of this paper. Additionally, we will continue to promote the application of EASI in more real industrial production scenarios.
> > > > >
> > > > > Again, sincerely thank you for your time and insights.

---

### Official Review · Reviewer_2SzS · 2024-07-12

**Soundness:** 3
**Presentation:** 3
**Contribution:** 2
**Rating:** 7
**Confidence:** 3

**Summary:**

The paper tackles the issue of finding correct parametrizations for simulators for robot tasks in order to close the sim-to-real gap. To optimize the simulation parameters, evolutionary strategies (ES) are employed in combination with a discriminator function (trained in a GAN setting). The study shows that the presented approach yields better results (running in whatever setting is considered "real") than the baseline approach Uniform Domain Randomization (UDR).

**Strengths:**

The paper tackles the important topic of closing sim-to-real gaps, which might have impat eve way beyond robots. The employed method appears rather straightforward, which is a big plus, since it means that either (i) it is in fact the natural method to use or (ii) the authors did such a good job sharing their perspective that the reader almost thought of the siolution him/herself. In either case, the paper shows empirical success by outperforming UDR.

The choice of domains seems appropriate and the study shed a light on the various behavorial properties of the presented approach. Albeit limited in space, the authors try to show limitations and problems of the approach as well. It is very important that the authors not only show the result but (with studies like that one behind Fig. 2, e.g.) they also provide some analysis on why things work.

I enjoyed that at multiple points throughout the paper, the current agenda is repeated and it is always clear what the authors want to show.

**Weaknesses:**

Some connections within the paper are very confusing. Alg. 1 is not properly referenced und thus not discussed deeply enough. Fig. 1 is not referneced properly and comes way too late within the paper. Various explanations on GANs are confusing as -- in this approach -- we do not have a generator in its own right but build a slightly different architecture. The plots in Fig. 2, e.g., are explained at various non-continuous locations and thus very hard to follow.

The study lacks some stronger comparison, but if the field does not have much to offer here at the moment, this is acceptable. I am missing some disucssion about the comparative use of compute. EASI seams much more involved than UDR here.

The limitation section is very good to have, but quite weak as limitation sections go.

Eq. 4 is unclear to me. Does it contain a typo?

Several formal issues persist, including the usage of abbreviated forms ("it's"), various references without a qualifier ("according to 4"), and missing spaces. I recommend thorough proofreading. I also find it to be more common to use "an RL problem" and "an MDP".

Some more practical illustrations on the impact of the method would have been nice, especially the real world experiments.

**Questions:**

Eq. 4 is unclear to me. Does it contain a typo?

**Limitations:**

The limitation section is very good to have, but quite weak as limitation sections go.

---

> ### Author Rebuttal · Authors · 2024-08-07
>
> # Response to Reviewer 2SzS
> `Q1. The study lacks some stronger comparison. `
>
> `A1.` In the **common response A4**, we introduce additional comparative experiments, including the FineTune and GARAT. In these experiments, EASI consistently demonstrates the best performance in transferring simulation trained policies to the target environment.
>
> `Q2. The comparative use of compute.`
>
> `A2.` On our platform, we use EASI for one of the most complex tasks, Go2. EASI completed the parameter search in less than 10 minutes and is able to adjust the simulation parameter distribution to an appropriate range. After determining the simulation parameter distribution, UDR is used for policy training or fine-tuning.
>
> While using UDR directly can save the time spent on parameter search, as mentioned in the paper, we often lack the knowledge to set a reasonable parameter range for UDR. Without specialized knowledge to adjust the UDR parameter range, the trained policy might not transfer well to the real environment. Thus, directly using UDR may require extensive expertise and trial and error, which can be time-consuming compared to using EASI. By using EASI, we can leverage a small amount of real-world demonstration to quickly determine the UDR simulation parameter range, avoiding trial and error and improving the performance of the sim-to-real policy.
>
> `Q3. Some connections within the paper are very confusing. `
>
> `A3.` We carefully reviewed the issues you've raised, and we consider your feedback both valuable and important. We will promptly adjust the layout of the images in the paper and provide additional explanations for Fig. 1, offering a more detailed introduction to its contents. For Fig. 2, we will modify the layout of the paper to better link the legend to the corresponding content. Regarding the unclear phrasing, we will also make corrections in the next version, ensuring that readers can more easily understand our intended message. Thank you for your insightful feedback!
>
> `Q4. Eq. 4 is unclear to me. Does it contain a typo?`
>
> `A4.` The objective of Eq. 4 is to find a discriminator D that can better distinguish between simulated data and real data. We have revised Eq. 4 to make it more rigorous:
>
> $$\mathop{\max}\limits_{D} [E_{d^{\mathcal{M}}(\mathbf{s},\mathbf{a},\mathbf{s}')}[D(\mathbf{s},\mathbf{a},\mathbf{s}')]-E_{d^{\mathcal{B}}{(\mathbf{s},\mathbf{a},\mathbf{s}')}}[D(\mathbf{s},\mathbf{a},\mathbf{s}')]].$$
>
> Thank you for raising these issues. We will promptly revise the paper and add additional explanations to make the formula expressions clearer and easier to understand.

---

> > ### Author Response · Authors · 2024-08-13
> >
> > Dear Reviewer 2SzS:
> >
> > Thank you once again for your insightful comments and helpful suggestions and we're thrilled for your recognition of our work. As the deadline for author-reviewer discussions is approaching,  if you have any further questions or concerns, please let us know. Thank you very much for your time.

---

> > > ### Comment · Reviewer_2SzS · 2024-08-14
> > >
> > > Your reactions to my notes were satisfactory, justifying the positive evaluation I gave originally.

---

### Official Review · Reviewer_rASM · 2024-07-13

**Soundness:** 2
**Presentation:** 2
**Contribution:** 2
**Rating:** 6
**Confidence:** 3

**Summary:**

The paper introduces Evolutionary Adversarial Simulator Identification (EASI), a novel sim-to-real method using a combination of Generative Adversarial Network (GAN) and Evolutionary Strategy to bridge the gap between simulation and reality. EASI optimizes simulator parameters by aligning state transitions between simulation and reality, enhancing policy transferability. The method is tested on various tasks, demonstrating superior performance and stability compared to existing sim-to-real algorithms through experiments.

**Strengths:**

(1) The paper presents a novel method to mitigate the sim-to-real performance gap for Reinforcement Learning algorithms, which has an influential impact on the practical meaning.


(2) It adopts the Generative Adversarial Learning paradigm, designs a fitness function as a discriminator to identify the parameter distribution, and trains an Evolution-based algorithm as a generator to search for the system parameters.

**Weaknesses:**

1) This paper lacks explanations on important technique details. Such as how the transition distribution is derived from the demonstration data.
2) Some parts of the paper's demonstration are confusing, for example, if Fig.3 reflects the training of proposed methods in real env or simulator env. If it is trained in a real env, it doesn’t make sense for sim-to-real transfer.
3) The paper discussed comprehensive related work, but very few baseline methods were compared regarding the performance in the paper, such as Grounded Action Transformation, etc.
4) The setting of the experiment is not as convincing as what is claimed when targeting the existing work’s problems. Such as line 42-43, other methods are hard to perform well in large-dimensional parameter space, this paper only conducted experiments on the environment with a maximum of 11 parameters, which might still be simple env.
5) The paper claimed `yes` in checklist 5, open-access of code, but it is not available anywhere in the paper, which is not aligned with the checking claims, and may hinder the reproducibility.

**Questions:**

1) How do you capture/model the state transition, since the sim-transitions will be used to measure the gap with the real-transitions by JS?
2) In algorithm lines 9-10, Do they require the same initial state s_k? If yes, how does the work make sure they are aligned? If not,  how do the authors measure the transition divergence? I assume the JS divergence was employed (line 157), it would require two distributions, how do the authors obtain/estimate the real-transitions distribution?
3) The baseline methods are only the Oracle and Uniform Domain Randomization(UDR), can authors incorporate the GAT branch methods, and Domain Adaptation methods as well?
4) I have concerns about the generalizability of the proposed method since for GAN-based methods, it is notoriously difficult to train, especially for high-dimensional feature space, and what is presented in the paper seems still a relatively simple environment.
5) Can authors explain If Fig.3 reflects the training of proposed methods in real env or simulator env? If it is trained in a real env, it doesn’t make sense for sim-to-real transfer.

**Limitations:**

1) The paper lacks proof of the high-dimensional transfer performance either empirically or theoretically.


2) And the work seems not able to easily adapt to new tasks, it requires training all over again (using the GAN paradigm) in a new real-world environment.

---

> ### Author Rebuttal · Authors · 2024-08-07
>
> # Response to Reviewer rASM
> `Q1. How do you capture/model the state transition`
>
> `A1.` In this paper, we draw on the training paradigm of GANs to estimate the distance between the simulated and real state transition distributions. According to GAN theory [1], Discriminator D and Generator G play the following two-player minimax game with value function $V(G,D)$:
>
> $$\mathop{min}\limits_{G} \mathop{max}\limits_{D}V(G,D)=E_{x\sim p_{data}(x)}[logD(x)]+E_{z\sim p_{z}(z)}[log(1-D(G(z)))].$$
>
> For any given fixed G, training D to maximize the
> quantity $V(G,D)$, we can proof that [1]:
>
> $$C(G)=\mathop{max}\limits_{D}V(G,D) =-log(4)+KL(p_{data}||\frac{p_{data}+p_{g}}{2})+KL(p_{g}||\frac{p_{data}+p_{g}}{2})$$
> $$C(G)=-log(4)+2\cdot JSD(p_{data}||p_{g}).$$
>
> $C(G)$ could view as the estimication of the Jensen–Shannon divergence between two distributions.
>
> We could leverage $C(G)$ to estimate the distance between state transition distributions. In each generation of evolution, **we only need to randomly sample $(s,a,s')$ tuples** from real state transition and simulated state transition to train the discriminator. Once the discriminator reaches its optimal, it can be used to estimate the distance between real and simulated state transition distributions.
>
> In EASI, we use a WGAN-style discriminator
>
> $$C(G) = \mathop{\max}\limits_{D} E_{d^{\mathcal{M}}(\mathbf{s},\mathbf{a},\mathbf{s}')}[D(\mathbf{s},\mathbf{a},\mathbf{s}')]-E_{d^{\mathcal{B}}{(\mathbf{s},\mathbf{a},\mathbf{s}')}}[D(\mathbf{s},\mathbf{a},\mathbf{s}')],$$
>
> to tackle the discriminator vanishing gradients problem, and for this instance, $C(G)$ approximate the Earth-Mover distance between two distributions [2].
>
> [1] Goodfellow, Ian, et al. "Generative adversarial nets." Advances in neural information processing systems 27 (2014).
> [2] Arjovsky, Martin, Soumith Chintala, and Léon Bottou. "Wasserstein generative adversarial networks." International conference on machine learning. PMLR, 2017.
>
>
> `Q2. Do they require the same initial state s_k?`
>
> `A2.` **EASI does not have specific requirements for the initial agent state $s_k$, nor does it require the alignment of trajectories and state transitions.** In fact, this is a key advantage of EASI. EASI uses discriminator to estimate the distance between the state transition distributions of simulation and demonstration. Unlike previous work, which required aligning sim and real trajectories and then comparing trajectory errors, EASI uses $(s,a,s')$ tuples to train a discriminator and estimate the distance of state transition distribution by the discriminator.
>
> `Q3. Can authors incorporate the GAT branch methods, and Domain Adaptation methods as well?`
>
> `A3.` In the **common response A4**, we introduced additional comparative experiments, including the FineTune and GARAT (a GAT branch method). In these experiments, EASI consistently demonstrated the best performance in transferring simulation trained policies to the target environment.
>
> `Q4. The generalizability of the proposed method`
>
> `A4.` Since EASI uses Evolutionary Strategies (ES) as the simulation parameter generator, it is much more stable in the training process compared to the original GAN algorithm and less sensitive to hyperparameters. In **common response A5**, we conducted a hyperparameter sensitivity analysis for EASI. The results showed that EASI has low sensitivity to hyperparameters, further demonstrating its stability.
>
> Additionally, We tested EASI's sim-to-real performance in the real world using the Unitree Go2 robot in **common response A2**. The task in this experiment is complex and challenging, and EASI is still able to exhibit the expected performance.
>
> `Q5. Can authors explain Fig.3`
>
> `A5.` The training was conducted entirely in the original sim environment, and we saved models from various stages of the training process. These saved models were then tested in the target environment (since it's sim-to-sim, model testing was very convenient). The x-axis represents the number of policy updates in the original sim environment during RL training, while the y-axis shows the performance of the saved model tested in the target environment.
>
> Through this figure, we want to show that EASI not only facilitates better transfer of policies trained in the original environment to the target environment, but also accelerating the training process because of the narrower Domain Randomization (DR) parameters range. We apologize for the confusion caused by Fig. 3. In future versions, we will include additional explanations to ensure Fig. 3 is better understood.
>
> `Q6. Code is not available anywhere in the paper.`
>
> `A6.`  We have uploaded all the [codes](https://anonymous.4open.science/r/EASI-0998/) and supplementary materials.

---

> > ### Comment · Reviewer_rASM · 2024-08-10
> > **Thanks for your clarification**
> >
> > Thanks for your rebuttal. My concerns have been resolved and decided to improve my score.

---

> > > ### Author Response · Authors · 2024-08-10
> > > **Response to Reviewer rASM**
> > >
> > > Dear Reviewer rASM:
> > >
> > > Thank you for your constructive feedback and great efforts in helping us improve our work.
> > > We will continually improve the paper based on your recommendations. If you have any further concerns about the paper, please contact us.
> > >
> > > Once again, thank you!

---

### Author Rebuttal · Authors · 2024-08-07

# Common Response
We sincerely appreciate the reviewers for their insightful feedback! In the following, we will first address the comments that are shared by multiple reviewers.

 `Q1. The significance of this work.`

 `A1.` Domain randomization (DR) has become one of the most popular sim-to-real algorithms due to its simplicity and adaptability for various robotic applications. EASI introduces a new method for sim-to-real that quickly identifies the parameter range for DR, **creating a more realistic simulation environment for policy training**. With its efficiency and ease of use, we believe EASI can serve as a valuable tool for  the sim-to-real processes.

`Q2. More complex real-world applications.`

`A2.` To test the performance of EASI in the real world, we conduct an experiment with the Unitree Go2 quadruped robot. The goal is to have the robot move in a canter gait based on speed commands from a remote controller. The inputs for this task include 57-dim proprioceptive observations, 4-dim estimated privileged observations, 29-dim historical encodings, 570-dim historical observations, and 7-dim control commands. The outputs are the target positions for the robot's joint angles. We search for 7 parameters: the PD parameters for the hip, thigh, and calf joints, as well as the body mass. Unlike the sim-to-sim experiment, we do not include ground friction in our search to allow the robot to adapt to various ground types it might encounter. We train the initial policy using UDR in simulation and use it to collect real-world data, sampling 2000 steps at a control frequency of 50Hz, which provides about **40 seconds of data**. Using these real-world samples, we then employ EASI for parameter search.

First, we demonstrate that after parameter search with EASI, the simulator became more realistic. In this experiment, we use the same initial policy and speed command (v=1.4m/s) to control the robot's movement in both sim and real environments. We plot the frequency spectrum of the robot's joint movements in **Fig.1 (a) of the Supp. PDF**. Despite being controlled by the same policy, there are significant differences in the joint movement frequencies between the origin simulator and real environments. After parameter search with EASI, the differences in the joint movement frequency spectrum are significantly reduced.

Subsequently, we test the real robot's ability to follow speed commands. The results in **Fig.1 (b)** show that policies trained with EASI parameters have better speed tracking capabilities compared to those trained with origin parameters. This also proves that policies trained in the EASI-optimized sim environment can perform better in the real world.

`Q3. If the number of parameters to search increases, what will happen to EASI?`

`A3.` To explore the performance of EASI as the number of simulation parameters increases, we conducted a sim-to-sim experiment with the Go2 environment, searching for 25 parameters. In the experiment, we searched for the PD parameters of all motors in the Go2 robot and the mass of the robot. The results of the parameter search are shown in **Fig. 2**

`Q4. More baseline methods to compare.`

`A4.` To better compare with existing sim-to-real algorithms, we add FineTune and GARAT [1] as baselines.

In FineTune, we use the same demonstration data as EASI to fine-tune the UDR policy, allowing the UDR policy to adapt to the real-world environment based on real demonstration.

GARAT, a member of the GAT algorithm family, utilizes a GAN method to train an action transition network. This network adjusts actions in simulation such that, after adjustment, the actions applied in the sim environment result in state transitions more similar to those in the real demos. During the training of GARAT, we also used the same demonstration data as EASI.

**Table 1 in Supp. PDF** shows the performance in target environment of different algorithms after training in simulation. **All the results demonstrate the advantages of the proposed method.**

[1] Desai, Siddharth, et al. An imitation from observation approach to transfer learning with dynamics mismatch. Advances in Neural Information Processing Systems 33 (2020): 3917-3929.

`Q5. Hyperparameter sensitivity analysis for EASI.`

`A5.` Since EASI uses Evolutionary Strategies (ES) as the simulation parameter generator, it is much more stable in the training process compared to the original GAN algorithm and less sensitive to hyperparameters. In EASI, we select two hyperparameters for sensitivity analysis in **Table 2**，the evaluation metric is the L2 error between the searched parameter and the target parameters.

We first analyze the **hyperparameter** $\mu/\lambda$. In each evolution, we test $\lambda$ individuals and select the best-performing $\mu$ elite individuals for recombination and mutation. Generally, a higher $\mu/\lambda$ value helps maintain diversity in the population but may slow down the evolution process. Conversely, if $\mu/\lambda$ is too low, a lack of diversity can lead to local optima. To enhance parameter search, we recommend using a larger $\mu/\lambda$ and increasing the number of evolution steps, which will improve EASI's performance but also increase computational time.

Next, we analyze another **hyperparameter epoch_disc**, which refers to the number of times the discriminator is trained before each generation of evolution. Insufficient training may prevent the discriminator from achieving optimal performance, while excessive training can cause overfitting. In our experiments, we found that varying epoch_disc had only a minor effect on the final search results.

`Q6. Code is not available.`

`A6.` We have uploaded all the [codes](https://anonymous.4open.science/r/EASI-0998/) and supplementary materials.

`Q7. Minor errors.`

`A7.` We have corrected minor errors in the paper and have incorporated the suggested changes.

Please let us know if there are any remaining questions!

---

### Decision · Program_Chairs · 2024-09-25

**Decision:**

Accept (poster)

**Comment:**

## Recommendation

I recommend "Accept (poster)".

## Summary

The manuscript presents a novel method for simulator identification for sim2real robot learning. The authors combine real-world demonstrations, Generative Adversarial Networks (GANs) and Evolutionary Strategies (ES) in order to learn a distribution of simulation parameters. In particular, using the demonstrations they learn a discriminator (using GANs), which is then used as an objective function to an ES process. The final outcome of the EASI algorithm is a distribution of simulation parameters, and then normal uniform domain randomization (UDR) is performed (although one can use the learnt distribution in other ways). Overall, the paper is well-written, solves an interesting problem, and extensive experimentation/discussion is provided (including real-world experiments).

## Concerns

My main concern (and thus recommending poster instead of spotlight) is the impact of the contribution of the proposed method. In particular, the work is very close (in spirit) to {1} ([8] in the original manuscript) and in my opinion {1} possess a stronger main contribution. To further expand on this, the method in {1} is a closed-loop algorithm that learns in the real-world while learning the "optimal" simulation parameters. This is extremely important as the underlying assumptions about the system are less strong. On the contrary, EASI only learns using a "static" dataset (static is in quotes, since nobody constrains us of using the EASI proposed GAN-ES pipeline online; but the authors did not do this), and then we need to use this parameters to learn a policy. Given the above comments/realization, I am not sure about the potential impact of the contribution.

I expect the authors to include a paragraph to discuss the above comments/limitation (possibly in Section 6).

## Verdict

Despite my concern, I recommend acceptance (poster) for the following reasons:

1) The paper is well-written and clearly articulates the goals, solutions and discusses the results/assumptions
2) The idea of using a discriminator to learn an objective function for a black-box optimization procedure is straightforward (not saying this in a bad way; I like straightforward!), and very interesting!
3) The authors' rebuttals were clear, concise and provided interesting discussions or new experiments. Overall, the authors' comments showcase deep understanding of the literature.
4) All reviewers were positive about the work.

## References

{1}: Chebotar, Y., Handa, A., Makoviychuk, V., Macklin, M., Issac, J., Ratliff, N. and Fox, D., 2019, May. Closing the sim-to-real loop: Adapting simulation randomization with real world experience. In 2019 International Conference on Robotics and Automation (ICRA) (pp. 8973-8979). IEEE.